# Making Instruction Finetuning Accessible to Non-English Languages: A Case Study on Swedish

**Oskar Holmström\***
Linköping University
oskar.holmstrom@liu.se

**Ehsan Doostmohammadi\***
Linköping University
ehsan.doostmohammadi@liu.se

## Abstract

In recent years, instruction-finetuned models have received increased attention due to their remarkable zero-shot and generalization capabilities. However, the widespread implementation of these models has been limited to the English language, largely due to the costs and challenges associated with creating instruction datasets. To overcome this, automatic instruction generation has been proposed as a resourceful alternative. We see this as an opportunity for the adoption of instruction finetuning for other languages. In this paper we explore the viability of instruction finetuning for Swedish. We translate a dataset of generated instructions from English to Swedish, using it to finetune both Swedish and non-Swedish models. Results indicate that the use of translated instructions significantly improves the models' zero-shot performance, even on unseen data, while staying competitive with strong baselines ten times in size. We see this paper is a first step and a proof of concept that instruction finetuning for Swedish is within reach, through resourceful means, and that there exist several directions for further improvements.

## 1 Introduction

The use of pretrained language models in natural language processing (NLP) is widespread, with finetuning or zero-shot approaches employed for various tasks. However, not all pretrained models exhibit strong zero-shot performance or are cost-effective to finetune for every new task. To overcome these limitations, instruction finetuning—finetuning on natural language processing tasks

that are described as instructions—has been demonstrated to enhance generalization to unseen NLP problems and tasks (Wei et al., 2022; Chung et al., 2022). Instruction finetuning, although beneficial, can be costly since it requires human annotation or feedback. To overcome this issue, automatic instruction generation has been demonstrated as a cost-effective alternative (Honovich et al., 2022; Wang et al., 2022a). While the benefits of automatic finetuning are substantial for English, which has abundant data resources, they are even more pronounced for languages with limited resources, such as Swedish.

In this paper, we explore how automatic methods for instruction finetuning can be extended to Swedish. The work is partly based on Unnatural Instructions (Honovich et al., 2022), a method of bootstrapping the instruction creation process. We use the generated instruction as a teacher to a Swedish student, where a translator module acts as an intermediary. The dataset is translated from English to Swedish and then used to finetune various Swedish and non-Swedish models to investigate the effectiveness of the proposed technique. The translations and models are evaluated using both human and automatic methods.

We find that the translated instructions generates a significant increase in zero-shot performance, even to unseen data. This paper is a first step, and a proof of concept, that instruction finetuning for Swedish is possible and that there exist several directions for further improvements.

## 2 Related Work

Language models have demonstrated the capability to solve tasks through following instructions in a zero-shot setting. However, their performance can be enhanced by finetuning on a diverse set of task-specific instruction data. This allows the model to adapt and generalize to new, unseen tasks, reducing the need for task-specific finetuning and enabling

---

*Equal contribution.

an off-the-shelf solution (Weller et al., 2020; Efrat and Levy, 2020; Mishra et al., 2022; Sanh et al., 2022; Chakrabarty et al., 2022; Gupta et al., 2022; Wang et al., 2022b). Manually procuring data for task specific finetuning can be costly. To mitigate this issue, researchers have explored automatically generating data (Schick and Schütze, 2021). Studies have shown that this alternative is highly effective, with performance that is only slightly behind that of large language models, which has been finetuned on manual data (Honovich et al., 2022; Wang et al., 2022a).

## 3 Automatic Instruction Finetuning for Swedish

We use two different instruction specific datasets and translate them to Swedish: The first one to finetune two models, and the second one as a held-out evaluation set. We evaluate the performance of the two models before and after instruction finetuning together with a strong GPT3 baseline to explore the usability of the automatically procured and translated instruction data.

The code, model checkpoints, and datasets used in the paper are made available[1].

### 3.1 Datasets

**UNNATURAL INSTRUCTIONS**   In our study, we use the core dataset of UNNATURAL INSTRUCTIONS (Honovich et al., 2022) as a base for training and testing the models. The dataset was generated by starting with 15 manually written samples as seed, and then incrementally adding more samples with OpenAI's `text-DaVinci-002`, using three seed examples to generate a fourth one at a time. Each sample contains the following parts: (1) the instruction, which is the definition of the task, e.g., "Find an answer to the mathematical problem."; (2) the input text which is a specific example in the instruction space, e.g., "A wheel has a circumference of 15 feet. What is its diameter in inches?"; (3) constraints, which specifies the restrictions of the expected answer, e.g., "The output should be a number, rounded off to 2 decimal places."; and (4) the output, which is the correct generation considering all the previous instructions and constraints.

The core set of the UNNATURAL INSTRUCTIONS dataset comprises 68,478 samples, which

we split into two sets: 100 samples for testing and the remainder for training. The top 10 tasks in the dataset belong to a broad set of categories, and are as follows: question answering, sentiment analysis, arithmetic, geometry, event ordering, fact verification, fill-in-the-blank, general math puzzles, identifying overlapping strings, and array manipulations and puzzles.

**NATURAL INSTRUCTIONS**   For evaluation of our models, we utilize a subset of the NATURAL INSTRUCTIONS dataset generated by human annotators (Mishra et al., 2022). The test set of this dataset comprises 12 tasks, and we randomly select 80 samples from each task to assess the models' performance using ROUGE-L, and a subset of randomly selected 5 sample per task for human evaluation. The tasks are question and answer generation with regards to different aspects of an incident. For example, "Jack played basketball after school, after which he was very tired. Question: How long did Jack play basketball?". The task descriptions and the expected generated answers are also longer on average, resulting in a more difficult test set compared to UNNATURAL INSTRUCTIONS. See Appendix A for an overview of the tasks.

**Automatic Translation**   For the automatic translation of the data into Swedish, we use off-the-shelf machine translation models. The UNNATURAL INSTRUCTIONS dataset is translated with DeepL[2] and the NATURAL INSTRUCTIONS dataset is translated with `GPT3-DaVinci-003`. To assess the quality of the translations, we conduct a human evaluation, which rates the translations from one to three based on two criteria: (1) grammaticality and naturalness, and (2) accuracy compared to the source text. 120 random samples were selected from the UNNATURAL INSTRUCTIONS dataset and 10 examples per task were selected from the NATURAL INSTRUCTIONS dataset. The evaluator rates the translations on a scale of 1 to 3, with 1 indicating significant errors, 2 indicating minor errors, and 3 indicating correct and natural translations. The results show an average rating of 2.75 for grammaticality and naturalness, and 2.41 for accuracy for the UNNATURAL INSTRUCTIONS dataset, and 2.83 and 2.55 for the NATURAL INSTRUCTIONS dataset.

**Perplexity Dataset**   In order to evaluate the quality of the models outlined in Section 3.2, we assess their perplexity. To guarantee that the generated

---

[1]`https://github.com/oskarholmstrom/sweinstruct`

[2]`https://www.deepl.com`

| Model | Perplexity |
|---|---|
| GPT-SW3 | 1.92 |
| GPT-SW3-UI | 2.66 |
| OPT | 2.79 |
| OPT-UI | 5.45 |
| GPT3-Curie | 2.41 |
| GPT3-Curie-I | 2.99 |
| GPT3-DaVinci | 1.91 |
| GPT3-DaVinci-I | 1.94 |

Table 1: Perplexity of all the models on the SVT dataset.

| Model | UI ROUGE-L | NI ROUGE-L |
|---|---|---|
| GPT-SW3 | 0.084 | 0.009 |
| GPT-SW3-UI | **0.542** | 0.124 |
| OPT | 0.071 | 0.006 |
| OPT-UI | 0.449 | 0.101 |
| GPT3-Curie | 0.060 | 0.030 |
| GPT3-Curie-I | 0.308 | 0.108 |
| GPT3-DaVinci | 0.083 | 0.026 |
| GPT3-DaVinci-I | 0.537 | **0.151** |

Table 2: ROUGE-L scores on UNNATURAL IN-STRUCTIONS (UI) and NATURAL INSTRUCTIONS (NI) test sets for all the models. The best results are in bold.

texts are of high quality, and to eliminate the possibility of evaluating the models on data that was part of their pretraining, we use a custom dataset comprised of current news articles from the Swedish national public television broadcaster, SVT[3]. Our dataset is made up of 357 articles covering a range of subjects, with an average length of 256 tokens per article. These articles were published between July 1st, 2022, and January 19th, 2023.

### 3.2 Models

**GPT-SW3** (Ekgren et al., 2022) is a GPT2-like (Radford et al.) model pretrained on the Nordic Pile, where 26% of the data is Swedish (Öhman et al., 2023). The model is available in different sizes, but as a proof of concept, we finetune and evaluate the model with 1.3B parameters.

**OPT** (Zhang et al., 2022) is a BartDecoder-like (Lewis et al., 2020) language model pretrained predominantly on English data. However, even models that are intended to be trained on English are exposed to other languages during pretraining due to language contamination (Blevins and Zettlemoyer, 2022). We choose to finetune OPT to gain a perspective on how the predominant language in the base model affects its instruction handling abilities. To allow for fair comparisons, we use the 1.3B parameter model. The model is openly available and is trained on publicly available data.

**GPT3** (Brown et al., 2020) is a proprietary, closed-source large language model. We use both the pre-trained and instruction tuned GPT3-DaVinci-003, which has 175B parameters, and GPT3-Curie-001, which has 6.7B parameters. We abbreviate the instruction finetuned versions with "-I".

---

[3]https://www.svt.se

### 3.3 Finetuning and Experimental Setup

Having the training and test data described in Section 3.1, we instruction finetune the GPT-SW3 and the OPT models described in Section 3.2. We call the new models GPT-SW3-UI and OPT-UI, as they are finetuned on the UNNATURAL INSTRUCTIONS (UI) dataset. The models are finetuned using a next token prediction objective for the output, given the description, input, and the constraints of the task. We do not calculate any loss on the three first parts of the sample and the output is attention-masked so that the models cannot gain information from the output. We finetune the models for 3 epochs, following (Honovich et al., 2022). As for the other hyperparameters of the model, we chose 2e-5 for learning rate, 0.1 for weight decay, and 0.1 for warm-up ratio with an AdamW optimizer (Loshchilov and Hutter, 2019). For generation during inference, we use beam search with beam size 4 and 0.75 for temperature.

## 4 Results

**Perplexity** We first start with a perplexity analysis to measure the language modelling quality of the models on unseen Swedish data using the dataset described in Section 3.1. When evaluating perplexity using token length normalization, tokenizers that generate sentences with more tokens are favored. However, this approach can be problematic in cross-lingual settings, where tokenizing unknown words may increase the number of tokens generated. To overcome this issue, we use character length normalization as it provides a fairer measurement of perplexity across languages (Liang et al., 2022; Yong et al., 2022). From the results in Table 1, we see that perplexity increases after in-

| Model | Natural | Related | Correct |
|---|---|---|---|
| GPT-SW3-UI | 2.39 | 2.38 | 1.84 |
| OPT-UI | 2.25 | 1.59 | 1.33 |
| GPT3-Curie-I | 2.45 | 2.38 | 1.76 |
| GPT3-DaVinci-I | 2.56 | 2.68 | 2.28 |

Table 3: Average score of generations from human evaluation of the models on the Natural Instructions dataset.

struction finetuning. It is especially pronounced for the smaller models, while the change is minimal for `GPT3-DaVinci`. `OPT`, not originally trained on Swedish, is capable of modelling Swedish but also seems to suffer the most from instruction finetuning.

**ROUGE-L** Following Wang et al. (2022a), Mishra et al. (2022), and Honovich et al. (2022), we use ROUGE-L to automatically measure the performance of the models. The results show that the non-instruction-finetuned models perform the worst for both datasets. With instruction finetuning, we observe an increase in the ROUGE-L scores, even a major increase for non-Swedish models. It is important to note that the `GPT3` models have *not* undergone instruction finetuning on our data. The results highlight the challenge of NATURAL INSTRUCTION tasks compared to UNNATURAL INSTRUCTIONS, partly due to task complexity and partially due to the length of the answers. For a breakdown of the results on all the tasks, refer to Table 4.

**Human Evaluation** We perform a human evaluation study to accompany the ROUGE-L score analysis of model generations quality. The evaluation was done on 5 randomly selected generations. Three annotators independently scored (1 = not true, 2 = somewhat true, 3 = true) the generations on three different criteria: (1) whether it is natural and grammatically correct; (2) whether it relates to the provided context; (3) if it is a correct answer for the given task. The instruction finetuned `DaVinci` model with 175B parameters produces more natural and correct outputs than the other models, while `GPT-SW3` and the instruction finetuned `Curie` model perform close to each other on all three criteria. The results are shown in Table 3.

## 5 Discussion

The results from the perplexity analysis show an increase for all models after instruction finetuning, even though the models become more capable at a broad set of tasks. It has been shown that perplexity does not correlate strongly with downstream task or prompting performance (Liang et al., 2022; Yong et al., 2022). However, when using automatically translated data, we need to be aware that noise in the translation process can affect the models' capabilities. The increase in perplexity could partly be explained by unwanted noise. A hypothesis for why we see a larger increase in perplexity for the `OPT` model than the `GPT-SW3` model is that stronger foundations in the target language makes the model more robust to translation errors.

The significant increase of ROUGE-L scores for all models, especially on the difficult NATURAL INSTRUCTIONS tasks, show that the models can become strong zero-shot generalizers with relatively little finetuning. Unsurprisingly, the stronger performance of `GPT-SW3` than `OPT` shows that a strong foundation in the target language is helpful.

While there are some issues with making direct comparisons with the baseline `GPT3` models, as we do not know what data they have seen during training, `GPT-SW3` outperforms the larger `Curie` model on both datasets and even the `DaVinci` model, two orders of magnitude larger, on the unnatural instructions test set. However, our human evaluation shows that there are still significant improvements that need to be made to reach parity with the largest model.

## 6 Conclusion and Future Work

Using automatically created instructions that have been translated to Swedish provides a significant increase in zero-shot performance when instruction finetuning a `GPT-SW3` and `OPT` model. The `GPT-SW3` model shows competitive performance against the hundred times larger instruction-tuned `GPT3-DaVinci` model. While the results are promising, this is still a work in progress. Human evaluations show that there is significant progress to be made, especially in giving correct answers to instructions. A possible path for performance gain is to study the effects of translation quality on models' performance. We also leave for future work how the automatically translated instruction finetuning interacts with increased model scale.

In conclusion, we find that instruction finetuning

for Swedish is not only within our reach, but it can be achieved with a completely automatic process that yields significant improvements on a broad set of tasks in a zero-shot setting.

## Acknowledgments

This work was partially supported by the Wallenberg AI, Autonomous Systems and Software Program (WASP) funded by the Knut and Alice Wallenberg Foundation. The computations were enabled by the Berzelius resources provided by the Knut and Alice Wallenberg Foundation at the National Supercomputer Center.

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

| Model | T1 | T2 | T3 | T4 | T5 | T6 | T7 | T8 | T9 | T10 | T11 | T12 |
|---|---|---|---|---|---|---|---|---|---|---|---|---|
| GPT-SW3 | 0.035 | 0.006 | 0.002 | 0.013 | 0.008 | 0.007 | 0.000 | 0.001 | 0.004 | 0.021 | 0.008 | 0.002 |
| GPT-SW3-UI | 0.033 | 0.204 | 0.085 | 0.106 | 0.067 | 0.156 | 0.286 | 0.080 | 0.125 | 0.083 | 0.080 | 0.187 |
| OPT | 0.060 | 0.002 | 0.004 | 0.0 | 0.003 | 0.001 | 0.001 | 0.003 | 0.000 | 0.000 | 0.001 | 0.000 |
| OPT-UI | 0.038 | 0.125 | 0.052 | 0.038 | 0.033 | 0.044 | 0.352 | 0.093 | 0.187 | 0.054 | 0.075 | 0.120 |
| GPT-Curie | 0.027 | 0.039 | 0.009 | 0.030 | 0.028 | 0.036 | 0.034 | 0.043 | 0.007 | 0.040 | 0.036 | 0.037 |
| GPT-Curie-I | 0.080 | 0.095 | 0.067 | 0.069 | 0.086 | 0.106 | 0.116 | 0.226 | 0.217 | 0.109 | 0.073 | 0.059 |
| GPT-DaVinci | 0.027 | 0.038 | 0.012 | 0.017 | 0.028 | 0.044 | 0.012 | 0.020 | 0.010 | 0.038 | 0.032 | 0.037 |
| GPT-DaVinci-I | 0.119 | 0.071 | 0.084 | 0.238 | 0.131 | 0.091 | 0.147 | 0.419 | 0.264 | 0.121 | 0.093 | 0.037 |

Table 4: A breakdown of ROUGE-L scores on the NATURAL INSTRUCTIONS (NI) test subsets for all the models.

## A   NATURAL INSTRUCTIONS Tasks

The following are a summary of the task descriptions in the NATURAL INSTRUCTIONS test set. Please refer to Mishra et al. (2022) for more information.

1. **Task 1**: Writing questions that require tracking entity references.

2. **Task 2**: Writing answers to questions involving multiple references to the same entity.

3. **Task 3**: Writing questions that ask about event duration, based on given sentences.

4. **Task 4**: Writing answers to questions that involve event duration, based on given sentences.

5. **Task 5**: Writing *implausible* answers to questions that involve event duration, based on given sentences.

6. **Task 6**: Writing questions that involve "transient vs. stationary" events[4].

7. **Task 7**: Writing answers to a questions that involve events that are stationary or transient.

8. **Task 8**: Writing *implausible* answers to a questions that involve "transient v. stationary" events.

9. **Task 9**: Writing questions that involve the ordering of events.

10. **Task 10**: Writing answers to questions about the events that may happen before or after certain events.

11. **Task 11**: Writing events that is *not likely* to happen after certain events or is *not likely* to have happened before it.

12. **Task 12**: Writing questions about when events happened.

A breakdown of how the models perform on each of these tasks are presented in Table 4.

[4]If an event will change over time, it is called transient and if not, it is called stationary.

## B   NATURAL INSTRUCTIONS Samples

The following are examples of inputs and outputs generated from the instruction finetuned GPT-SW3 model.

### B.1   Positive examples

Uppmaning: Ställ en fråga om "händelseordning" baserat på den givna meningen. Definition: I denna uppgift, baserat på en given mening, ber vi dig att skriva en fråga som involverar ordning av händelser, dvs. förståelsen av hur händelser vanligtvis ordnas. Till exempel "tjäna pengar" brukar vanligtvis förekomma före "spendera pengar"

Saker att undvika: Skapa inte frågor som har explicita nämningar av svar i text. Istället måste det antydas från det som ges. Med andra ord vill vi att du använder "instinkt" eller "sunt förnuft".

Betoning & Försiktighet: De skrivna frågorna kräver inte att de har ett enda korrekt svar.

Input: Mening: När våra bilar bränner bensin i sina motorer, omvandlas det till rörelse och värme.

Output: Hur omvandlas bensin till rörelse och värme?

Uppmaning: Ställ en fråga om "absolut tidspunkt" för händelser baserat på den givna meningen. Definition: I denna uppgift, baserat på den givna indata, ber vi dig att skriva en fråga som involverar "absolut tidspunkt" för händelser, vilket definieras som förståelse för när händelser vanligtvis händer. Till exempel "att gå till skolan" händer vanligtvis under dagen (inte klockan 2 på natten).

Saker att undvika: Skapa inte frågor som har explicita nämningar av svar i text. Istället måste det antydas från det som ges. Med andra ord vill vi att du använder "instinkt" eller "sunt förnuft".

Betoning & Försiktighet: De skrivna frågorna kräver inte ett enda korrekt svar.

Input: Mening: Lennon anklagar sin far för att lämna honom igen, och sedan lämnar han, efter att ha berättat för sin far att han inte kommer att leva med honom längre.

Output: När lämnade Lennon sin far?

### B.2   Negative examples

Uppmaning: Ge ett negativt/felaktigt svar på frågan om "händelseordning". Definition: I denna uppgift ber vi dig att skriva ett osannolikt svar på en fråga som involverar ordning av händelser, dvs. förståelsen av hur händelser vanligtvis ordnas. Till exempel "tjäna pengar" brukar vanligtvis förekomma före "spendera pengar"

Saker att undvika: -

Betoning & Försiktighet: Även om det finns flera felaktiga svar, behöver vi bara ett felaktigt svar. Försök att hålla ditt "svar" så enkelt som möjligt. Kortfattat och enkelt "svar" föredras framför de komplexa och långa.

Mening: Vid nuvarande användningshastigheter kommer kol att räcka i cirka 300 år. Fråga: Vad gjorde människor efter att ha insett att kol inte kommer att räcka för evigt?

Output: Människor insåg att kol inte kommer att räcka för evigt.

Uppmaning: Ställ en fråga om "händelsens varaktighet" baserat på den givna meningen. Definition: I denna uppgift ber vi dig att skriva en fråga som involverar "händelsens varaktighet", baserat på en given mening. Här definieras händelsens varaktighet som förståelsen av hur långa händelser vanligtvis varar. Till exempel tar "borsta tänderna" vanligtvis några minuter.

Saker att undvika: Skapa inte frågor som har explicita nämningar av svar i text. Istället måste det antydas från det som ges. Med andra ord vill vi att du använder "instinkt" eller "sunt förnuft".

Betoning & Försiktighet: De skrivna frågorna kräver inte att de har ett enda korrekt svar.

Mening: Trots sin snabba klättring uppför juridiska stegen har Bailey alltid hittat tid att hjälpa till i de orsaker som han känner starkt för.

Output: Varaktighet?

## C   UNNATURAL INSTRUCTIONS Samples

The following are examples of inputs and outputs generated from the instruction finetuned GPT-SW3 model.

### C.1   Positive examples

Input: Du ska beräkna medelvärdet av tre givna tal med hjälp av följande operatorer: addition, subtraktion, multiplikation och division. Svaret ska vara korrekt med en decimal. 15, 12, 18.

Begräningsning: Ingen.

Output: Genomsnittet av de tre givna talen är 15

Input: I den här uppgiften ombeds du att komplettera en kort berättelse genom att fylla i luckorna med ett ord. Den givna meningen är en antydan om vilken typ av ord som ska användas för att fylla i tomrummet. Jag kände mig ___ efter min långa dag på jobbet.

Begräningsning: Output måste vara ett adjektiv.

Output: Trött

Input: I den här uppgiften måste du vända en given sträng. Hello World!

Begräningsning: Ingen.

Model output: !dlrow olleH

## C.2 Negative examples

Input: Skriv en funktion som ger en resplan med minsta möjliga kostnad, med en lista över flygbiljettpriser och en annan lista med motsvarande antal mellanlandningar för varje flygning. Anta att det bara finns ett anslutningsflyg från en stad till en annan. Priser = [200, 400, 350], mellanlandningar = [0, 2, 1].

Begräningsning: Ingen.

Output: Den minsta möjliga kostnaden är 20.

Output: Om du antar att du använder en övervakad inlärningsalgoritm måste din klassificerare tränas på märkta bilder av människors ansikten. För att kunna göra detta behöver du ett dataset med märkta bilder av människors ansikten. Det finns många sådana dataset tillgängliga på nätet, t.ex. datasetet Labeled Faces in the Wild.

Model output: En möjlig klassificerare skulle kunna vara en funktion som tar in människors ansikten tillsammans med etiketterna "man" och "kvinna" och returnerar en funktion som tar in människors ansikten tillsammans med etiketterna "man" och returnerar en funktion som tar in människors ansikten tillsammans med etiketterna "man" och returnerar en funktion som tar in människors ansikten tillsammans med etiketterna "man" och returnerar en funktion som tar in människors ansikten tillsammans med etiketterna "man" och returnerar en funktion som tar in människors ansikten tillsammans med etiketterna "man" och returnerar en funktion som tar in människors ansikten tillsammans med etiketterna "man" och

Input: Du får en lista med strängar. Uppgiften är att återge antalet unika strängar i listan som har en längd som är större än 1 och där alla tecken i strängen är små bokstäver. ['hello', 'world', 'my', 'name', 'is'].

Begräningsning: Det ska vara ett heltal som representerar antalet unika strängar.

Output: [No model answer]

## C.3 Noisy translation examples

Input: Du får en lista med ord. Din uppgift är att gruppera de givna orden i två grupper: de som rimmar och de som inte rimmar. katt, hatt, sat, fladdermus, mat.

Begräningsning: Inga.

Output: katt, fladdermus, mat.

Input: Du måste hitta den tjockaste personen i följande lista över personer. Tim, 1,80 meter lång, väger 140 pund. Sarah, 1,5 meter lång, väger 160 pund. Mike, 5 fot och 2 tum lång, väger 180 pund.

Begräningsning: Utfallet ska vara en mening som beskriver vem som är tyngst och hur mycket han eller hon väger.

Output: Sarah väger 160 pund.

## C.4 Example of noise in gold data

Input: Du får en uppsättning heltal. Din uppgift är att bestämma medianvärdet i den givna mängden. Medianen är det mittersta värdet när värdena i en mängd listas i antingen stigande eller fallande ordning. Om det finns ett jämnt antal värden bestäms medianen genom att man tar medelvärdet av de två centrala värdena. 3, 7, 1, 9.

Begräningsning: Ingen

Gold output: Medianvärdet är 7.

Model output: Medianen är 5.