# OpenReview forum: "Making Instruction Finetuning Accessible to Non-English Languages: A Case Study on Swedish Models"
_NoDaLiDa/2023/Conference — NoDaLiDa 2023_

### Official Review · Reviewer_Ur7B · 2023-03-08
**Good short paper on instruction finetuning for Swedish GPT-models.**

**Rating:** 8
**Confidence:** 5

**Review:**

This paper discusses instruction finetuning of generative language models in languages other than English, and presents an experiment with translating English instruction data to Swedish and using this data to finetune a Swedish generative language model (GPT-SW3). The Swedish model is compared to two different English-based models (OPT and GPT-3), and the results indicate that instruction finetuned GPT-SW3 performs competitive to the 100 times larger GPT-3.

The paper is clearly written with good coverage of related work and relevant models, and presents a concise and convincing case. Both the results and the translated data will be interesting and useful for the community.

It is of course difficult to compare models without having the ability to control the entire training and finetuning setup. This is noted in the paper, but I would suggest to add a bit more discussion about the differences between the models, and about the challenges with making comparisons between open vs closed models. It is also not clear why only the relatively small version of GPT-SW3 was used (1.3B). It would have been interesting to see if using one of the larger versions of that model would have improved the results even further (as could perhaps be expected).

I could only find one minor error in the text: the caption for Table 1 is not correct (seems to have been copy pasted from Table 2).

**Paper Type:**

Short paper

---

### Official Review · Reviewer_WCNE · 2023-03-12
**Work in progress results with a potentially significant practical impact**

**Rating:** 8
**Confidence:** 4

**Review:**

The paper presents a work in progress research on instruction-finetuning large language models for Swedish. Essentially, this work aims at improving zero-shot performance of the Swedish GPT-SW3 model while comparing the results with instruction finetuned versions of the OPT and GPT-3 models that are pre-trained primarily for English.

This is a rather practice-oriented research. The authors explore a relatively low-cost finetuning scenario by machine translating a synthetic training dataset from English to Swedish, finetuning the GPT-SW3 and OPT models for 3 epochs, and evaluating the finetuned models in terms of perplexity and ROUGE-L score. Also a small scale human evaluation is conducted.

The results are very promising and inspiring - the finetuned 1.3B parameter GPT-SW3 shows competitive performnce (for Swedish) against the 175B parameter GPT-3-DaVinci model. A cost-effective result, although the human evaluation shows that there is still a significant space for improvements.

Overall, the paper is well written, and it presents original work for Swedish, which can be adapted for other languages as well.

Some minor issues:
- The concept of instruction finetuning is explained only in Section 3.1 when introducing the dataset.
- Part 1 of the sample in Section 3.1 is not related to parts 2 and 3; a consistent sample would be better.
- The subsection of 3.1 on automatic translation does not contain a reference to the NMT model used.
- The caption of Table 1 doesn't correspond to its content.
- The lenghty Appendices B and C are accessible only to Scandinavian readers.

**Paper Type:**

Short paper

---

### Official Review · Reviewer_zMrU · 2023-03-14
**Initial case study of instruction fine-tuning for Swedish**

**Rating:** 8
**Confidence:** 4

**Review:**

This short paper presents an initial case study of instruction fine-tuning for Swedish using a Swedish generative language model (GPT-SW3 1.3B) and an English model (OPT 1.3B) as well as "pure" LM and instruction-tuned 6.7B and 175B variants of the original GPT-3. Fine-tuning and evaluation use machine translated versions of the previously introduced Unnatural Instructions (UI) and Natural Instructions (NI) resources. The authors report the results of a brief manual evaluation of MT quality and assess the pure LM and instruction-tuned versions in terms of perplexity on a small novel dataset of recent news articles (SVT) as well as ROUGE-L on the machine translated UI (held-out portion) and NI datasets. Finally, a small-scale human evaluation on 5 generations for NI is included to assess whether they are natural, related, and correct for the prompt.

Results show that the GPT-SW3 and in particular the OPT model suffer notably in terms of perplexity from the fine-tuning, while all instruction-tuned variants are (as expected) better at the UI and NI tasks. The Swedish model also achieves lower perplexity as well as higher ROUGE-L than the English OPT in both variants, while the largest instruction-tuned GPT3 achieves the highest ROUGE-L for NI and nearly matches the Swedish model on UI. Human evaluation finds the fine-tuned Swedish model and the 6.7B GPT3 to have comparable quality, while the largest GPT3 gets the highest overall score.

While somewhat preliminary in its conclusions and limited in its scope, the manuscript addresses a highly timely question likely to be of interest to many in the audience, namely whether smaller language-specific models can be fine-tuned to match some of the remarkable zero- and few-shot capabilities of very large English models. Its results are promising although not conclusive; as the authors note, the manuscript presents work in progress, as is appropriate for a short paper. Some detailed comments:

* The title (“… Case Study on Swedish Models”) appears to suggest several Swedish models are considered, but only one (GPT-SW3 1.3B) is used in the study. Perhaps consider just “Case Study on Swedish”?
* The only information provided regarding the MT model or its application is “we use an off-the-shelf machine translation model.” Please at least identify the model and preferably also the specific implementation used to perform the translation.
* The authors commit to releasing the translated versions of the UI and NI resources, but provide no information about the availability of the SVT dataset used in the perplexity evaluation. To allow replication (even in principle), it would be positive if the authors could provide means to access or recreate this dataset. (If copyright issues prevent republication, e.g. article URLs could potentially be used to identify the data.)
* Table 1 caption is wrong (repeats start of Table 2 caption)

**Paper Type:**

Short paper

---

### Decision · Program_Chairs · 2023-03-17

Accept